# Ethnic differences in healthcare trust and patient satisfaction in England: A cross-sectional survey

Steven David Pickering[1,2]*, Martin Ejnar Hansen[3], Han Dorussen[4], Jason Reifler[5], Thomas Scotto[6], Yosuke Sunahara[7], Dorothy Yen[2]

1 Department of Political Science, University of Amsterdam, Amsterdam, The Netherlands , 2 Brunel Business School, Brunel University of London, London, United Kingdom, 3 Department of Social and Political Sciences, Brunel University of London, London, United Kingdom, 4 Department of Government, University of Essex, Colchester, United Kingdom, 5 Department of Politics and International Relations, University of Southampton, Southampton, United Kingdom, 6 School of Government and Public Policy, University of Strathclyde, Glasgow, United Kingdom, 7 Graduate School of Law / Division of Law and Political Science, Kobe University, Kobe, Japan

* s.d.pickering@uva.nl

## Abstract

Trust in healthcare is an important component of patient experience, yet ethnic minority patients in England often report lower trust and satisfaction with the NHS than White patients. We examine these differences using data from a cross-sectional survey of 1,111 adults living in England, conducted in April–May 2023. Respondents rated their agreement with 18 statements covering five domains of healthcare experience: communication, ethical treatment, perceived competence, trust in providers, and perceived discrimination. We estimate a series of linear regression models comparing responses between ethnic minority and White participants, both before and after adjustment for demographic and attitudinal factors. Ethnic minority respondents reported significantly less positive perceptions on several items, particularly those relating to ethical behaviour, personal care, trust in front-line providers and perceived discrimination. They were less likely than White respondents to agree that the NHS behaves ethically, cares for patients, or goes out of its way to help people, and reported lower trust in GPs and hospital doctors and nurses. The largest differences were observed for perceptions that NHS care quality depends on ethnic background or country of origin. By contrast, we found no ethnic differences in perceptions of NHS honesty, promise-keeping, overall competence, or trust in NHS leadership or management. Overall, the findings indicate that ethnic differences in trust and satisfaction with the NHS are concentrated in relational and experiential aspects of care rather than in assessments of technical competence. Addressing these disparities may be important for improving equity and strengthening trust in healthcare services.

## Introduction

Patient trust and satisfaction are widely recognized as important elements of healthcare quality and patient experience [1]. Trust in healthcare providers is central to

**Data availability statement:** Replication data and code are available from the Harvard Dataverse, at: https://doi.org/10.7910/DVN/U099DY.

**Funding:** This research was funded by the Japan Society for the Promotion of Science (JSPS, grant JPJSJRP 20211704) and the UK Research and Innovation's Economic and Social Research Council (UKRI-ESRC, grant ES/W011913/1). The funders had no role in study design, data collection and analysis, decision to publish, or preparation of the manuscript.

**Competing interests:** The authors have declared that no competing interests exist.

the functioning of health systems, as high levels of trust in providers facilitate better communication, adherence to medical advice, and positive health outcomes [2]. Conversely, lack of trust can undermine the therapeutic relationship and deter individuals from seeking care. Public satisfaction with healthcare services is closely monitored in England, as it is viewed as an important indicator by providers, policymakers, and the public [3]. Over the past decade, numerous studies have documented disparities in patient experience within the English NHS [4–6]. Ethnic minority patients as a group often report lower satisfaction with care and more negative experiences compared to White British patients [3]. A substantial body of research finds that ethnic minority patients report less favourable experiences across multiple dimensions of care [7–9]. For example, the large English General Practice Patient Survey has identified an "ethnicity effect" in patient experiences [10,11], and one recent analysis found significant gaps in primary care satisfaction between ethnic minority and White patients even after adjusting for socioeconomic factors [10]. Similar patterns appear in healthcare communications: minority patients report less favourable doctor-patient communication than Whites [7].

These disparities are not limited to primary care. Hospital inpatient surveys also reveal ethnic gaps. Patients of Pakistani and Bangladeshi background have often been among the least satisfied with hospital care, whereas White British or Irish patients report the highest satisfaction [12]. A large national Cancer Patient Experience Survey likewise found that ethnic minority cancer patients reported fewer positive experiences and lower overall satisfaction with care than White patients [13], although some cross-national studies have not consistently found this effect [14]. Beyond satisfaction, there is also evidence of lower public trust in the NHS among ethnic minority communities. Recent research highlights that people from minority ethnic groups tend to trust the NHS less than their White counterparts [15]. Several factors may be associated with this trust gap. Long-standing structural inequalities in healthcare, such as differences in access, treatment quality, and outcomes, are frequently cited as potential contributors, particularly among those who feel underserved or discriminated against [4–6]. Broader social and political influences can also affect trust; for instance, people's socio-political attitudes and general social trust correlate with their views of the NHS [16].

Despite the recognition of ethnic inequalities in healthcare experiences and trust, much of the existing research has examined trust and satisfaction at an aggregate level, such as overall trust in the NHS [15]. Less is known about which specific aspects of healthcare experience contribute to lower trust and satisfaction among ethnic minority patients. For example, do minority and White patients differ in how honestly they feel the NHS communicates, how ethically it behaves, or how competent it is in delivering care? Are there particular points of interaction, such as trust in GPs versus hospital staff, where the ethnic gap is greatest? More granular evidence on these questions is therefore needed, especially given the well-documented gaps in large-scale health data on ethnicity [17,18].

This study aims to address these gaps by examining multiple dimensions of trust and satisfaction with the NHS and comparing responses of ethnic minority and White

participants. By focusing on specific domains, such as communication, ethical treatment, competence, and perceived discrimination, we seek to identify where the largest perception gaps lie. Importantly, our focus is on patient-reported experience: how people feel about their interactions with, and trust in, the health system. This patient-centred lens aligns with the growing emphasis on patient experience in evaluating healthcare quality and equity. The findings are intended to provide evidence that may help inform discussion about how ethnic minority patients experience NHS care. If certain aspects (for example, feeling cared for or trusting one's GP) are particularly weak for minority respondents, these areas warrant closer attention in future research and practice.

## Methods

We conducted a cross-sectional survey study to examine healthcare trust and satisfaction across ethnic groups in England. Data were obtained from two waves of an online panel survey administered by YouGov from 19−21 April and 15−17 May 2023. Participants were recruited from YouGov's online panel of adults residing in England, using quota-based sampling designed to approximate the demographic composition of the adult population. The analyses presented here are not intended to produce nationally representative estimates, but to examine associations within the sample. The combined survey waves yielded a total analytic sample of 1,111 respondents, aged 18–88 years. All data were collected anonymously; no personally identifiable information was made available to the researchers. Participants provided informed opt-in consent electronically prior to completing the survey. All participants were adults. Ethical approval was granted by the Brunel University of London Ethics Committee (reference 35290-LR-Jan/2022-37313-1). All statistical analyses were conducted using R (version 4.5.3).

The survey included 18 statements assessing respondents' attitudes toward, and experiences with, the NHS. These statements are presented in Supplementary S1 Table and cover five key domains of patient experience: communication (statements 1–3), competence (statements 4–6), ethical care (statements 7–11), trust in healthcare staff (statements 12–15), and perceived structural discrimination (statements 16–18). For each statement, respondents indicated their level of agreement on a 7-point Likert scale from 1 ("strongly disagree") to 7 ("strongly agree"). For most items, higher scores indicate more positive evaluations of the NHS. For the discrimination items, however, higher scores indicate stronger perceptions of inequity or unusual treatment.

We also collected demographic and attitudinal variables as independent variables. Ethnicity was the primary variable of interest. Respondents self-identified their ethnicity via the online questionnaire. The survey used YouGov's standard ethnicity classification, which is based on the UK Office for National Statistics' (ONS) recommendations for gathering ethnicity data specifically in England [19]. This categorisation distinguishes White ethnic backgrounds from a range of mixed, Asian, Black, Middle Eastern and other ethnic identities. The category is chosen by the respondent. For the purposes of analysis, we constructed a binary ethnic minority indicator. Following ONS recommended practice, respondents selecting any White category (English/ Welsh/ Scottish/ Northern Irish/ British, Irish, Gypsy or Irish Traveller, or Any other White background) were coded as White (and therefore as 0 on our ethnic minority binary). Respondents selecting any non-White category, including mixed ethnic backgrounds, South Asian, East Asian, Black African or Caribbean, Arab or other ethnic identities, were coded as belonging to an ethnic minority group. Respondents who selected "prefer not to say" were excluded from the analysis.

For the analysis, we conducted a series of linear regression models to estimate the association between respondent ethnicity and each of the 18 satisfaction/trust statements. First, for each outcome statement, we fit an unadjusted model with ethnicity as the sole predictor. This model simply compared the mean response between ethnic minority and White respondents. Next, we fit an adjusted model for each outcome, including ethnicity and all other independent variables (gender, age, education, political orientation, and general trust) as predictors. By comparing unadjusted and adjusted results, we could assess whether observed ethnic differences were attenuated after accounting for these characteristics, rather than attributing them solely to ethnicity.

Each regression yielded a coefficient for the ethnic minority indicator, representing the difference (minority minus White) in agreement scores on the 7-point scale for each statement. A negative coefficient indicates that ethnic minority respondents, on average, reported lower agreement than White respondents (i.e., less positive evaluations of the NHS on that item). We calculated 95% confidence intervals for all coefficients. An ethnic difference was considered statistically significant when the confidence interval did not include zero. Although the outcomes are measured on 7-point Likert scales, we treat them as approximately continuous and estimate linear regression models. This approach is common in health and social science research and is supported by methodological evidence showing that parametric analyses are robust for Likert-type items with five or more response categories [20,21].

The survey data were supplied with post-stratification weights provided by YouGov to align the sample with population demographic benchmarks. Our primary analyses are based on unweighted data. As a robustness check, we re-estimated the adjusted models using the survey weights. The weighted estimates were substantively identical to the unweighted results: coefficients and confidence intervals differed only at the third decimal place, and the pattern of statistical significance was unchanged. Accordingly, we report the unweighted estimates in the main text for simplicity. Weighted models are available in the replication code.

We assessed whether the assumptions of the linear regression models were met using standard diagnostic procedures. Linearity was evaluated by inspecting residual-versus-fitted plots, which did not indicate substantial departures from linearity. The distribution of residuals was examined using normal Q–Q plots; minor deviations from normality were observed in the tails, which are expected given the discrete nature of the outcome variables, but were not substantial and are unlikely to affect inference given the sample size. Multicollinearity was assessed using variance inflation factors (VIF), all of which were low (maximum VIF = 1.18), indicating no evidence of problematic collinearity among predictors. Overall, diagnostic checks suggested that the model assumptions were adequately satisfied.

## Results

We begin by describing the demographic characteristics of the analytic sample before presenting item-level descriptive statistics and regression results. In the final analytic sample (N = 1,111), 98 respondents (8.82%) identified as belonging to an ethnic minority group and 1,013 (91.18%) identified as White (see Table 1). Supplementary S2 Table compares the demographic characteristics of the ethnic minority subsample with the 2021 Census non-White population in England. The comparison indicates that while the ethnic minority subsample is somewhat younger than the population benchmark, other characteristics are broadly comparable on key demographic dimensions. We acknowledge that this broad categorization masks important heterogeneity within minority ethnic groups; however, the sample size did not permit reliable disaggregation by specific ethnicity. The proportion of ethnic minority respondents in our sample is below the national estimate for England (18.3%) [22]. Under-representation of minority groups in surveys is a well-documented challenge [23,24] reflecting structural barriers to participation, and a range of methodological strategies have been proposed to address it [25,26]. Although the number of ethnic minority respondents limits the precision of some estimates, it is sufficient for detecting moderate group differences. Accordingly, non-significant findings should not be interpreted as evidence of the absence of ethnic differences. Ethnic minority respondents in the sample were, on average, younger than White respondents, reflecting known demographic patterns in England. All adjusted analyses account for age, gender, and educational differences.

Gender was coded as female or male (no respondents identified outside the binary in this survey). In the analytic sample, 54.91% of respondents were women and 45.09% were men. We included a binary indicator for higher education (university degree) to account for socioeconomic factors that might influence perceptions of the NHS. Respondents also placed themselves on a left-right ideological scale (0 = far left to 10 = far right), since political orientation can correlate with views of the NHS [16]. Age was included as a continuous covariate, given well-documented age differences in healthcare experiences and population composition in England. To capture baseline trustfulness, we included a measure of generalized interpersonal trust (where 1 indicated strong disagreement that other people could be trusted, and 7 meant strong

**Table 1. Descriptive statistics.**

| Characteristic | Value |
|---|---|
| N | 1111 |
| **Age** | |
| Min age | 18 |
| Max age | 88 |
| Mean age | 50.55 |
| SD age | 17.29 |
| Age: 18–24 | 103 (9.27%) |
| Age: 25–34 | 121 (10.89%) |
| Age: 35–49 | 308 (27.72%) |
| Age: 50–64 | 271 (24.39%) |
| Age: 65+ | 308 (27.72%) |
| **Gender** | |
| Women | 610 (54.91%) |
| Men | 501 (45.09%) |
| **Ethnicity** | |
| Ethnic minority | 98 (8.82%) |
| White | 1013 (91.18%) |
| **Education** | |
| Education: less than university | 702 (63.19%) |
| Education: university | 409 (36.81%) |
| **Ideology** | |
| Ideology: left | 350 (31.5%) |
| Ideology: centre | 395 (35.55%) |
| Ideology: right | 353 (31.77%) |
| Ideology: prefer not to say | 13 (1.17%) |
| Ideology (0–10) | Mean: 5.03; SD: 2.09 |
| **Trust** | |
| General trust (1–7) | Mean: 3.66; SD: 1.59 |

agreement), as individuals who are more trusting in general may also express greater trust in institutions [16]. Additional information (e.g., region) was collected but is not central to the present analysis.

Table 2 reports descriptive mean agreement scores and standard deviations for all survey items, stratified by ethnicity, to provide context for the regression results presented below.

Table 3 presents the results of the unadjusted and adjusted linear regression models estimating the association between ethnicity and each of the 18 NHS items. The coefficients represent the mean difference in agreement scores between ethnic minority and White respondents (ethnic minority minus White), with 95% confidence intervals.

Unadjusted models are estimated on the full analytic sample (N = 1,111). Adjusted models include gender, educational attainment (university degree), left–right political orientation, generalized interpersonal trust, and age, and are estimated on a reduced sample (N = 1,096) due to missing data on covariates. Adjusted models exhibit modest explanatory power, with $R^2$ values ranging from 0.019 to 0.090 and adjusted $R^2$ values ranging from 0.014 to 0.085 across outcomes (full details in Supplementary S3 Table). 95% confidence intervals are shown. * $p < 0.05$; ** $p < 0.01$; *** $p < 0.001$.

Overall, the results indicate a consistent pattern of ethnic differences across several domains of healthcare experience. While many items show no statistically significant differences between ethnic minority and White respondents, disparities

**Table 2. Results of the survey scores on trust in and satisfaction with the NHS in England stratified by minority and non-minority ethnic groups.**

| Item | Statement | White | | Ethnic Minority | | Men | | Women | | Less than Uni | | University | |
|---|---|---|---|---|---|---|---|---|---|---|---|---|---|
| | | Mean | SD | Mean | SD | Mean | SD | Mean | SD | Mean | SD | Mean | SD |
| | *Communication* | | | | | | | | | | | | |
| 1 | I believe the NHS is always honest in its communications. | 4.39 | 1.52 | 4.21 | 1.38 | 4.43 | 1.58 | 4.33 | 1.45 | 4.34 | 1.53 | 4.43 | 1.47 |
| 2 | The NHS is frank in its communications to people. | 4.35 | 1.46 | 4.2 | 1.55 | 4.36 | 1.5 | 4.32 | 1.43 | 4.28 | 1.45 | 4.45 | 1.48 |
| 3 | Information on important medical care issues is communicated openly by the NHS. | 4.6 | 1.43 | 4.2 | 1.45 | 4.63 | 1.47 | 4.5 | 1.4 | 4.51 | 1.4 | 4.65 | 1.48 |
| | *Competence* | | | | | | | | | | | | |
| 4 | The NHS is competent in providing a national healthcare service to the people of the UK. | 4.37 | 1.69 | 4.23 | 1.5 | 4.44 | 1.72 | 4.3 | 1.64 | 4.33 | 1.67 | 4.41 | 1.69 |
| 5 | The NHS knows how to supply the healthcare needs of our country. | 4.55 | 1.64 | 4.4 | 1.65 | 4.52 | 1.65 | 4.55 | 1.63 | 4.51 | 1.62 | 4.57 | 1.67 |
| 6 | I feel confident that the NHS can support and pro-vide quality healthcare to the people in the UK. | 4.2 | 1.7 | 4.2 | 1.49 | 4.29 | 1.7 | 4.14 | 1.67 | 4.22 | 1.71 | 4.17 | 1.64 |
| | *Ethical care* | | | | | | | | | | | | |
| 7 | Promises made by the NHS are always delivered. | 3.68 | 1.4 | 3.7 | 1.51 | 3.78 | 1.46 | 3.6 | 1.36 | 3.7 | 1.4 | 3.65 | 1.42 |
| 8 | I can count on the NHS to do things in an ethical manner. | 4.79 | 1.4 | 4.1 | 1.43 | 4.75 | 1.46 | 4.71 | 1.38 | 4.66 | 1.42 | 4.84 | 1.4 |
| 9 | The NHS always put people's interests before its own. | 4.21 | 1.56 | 4.08 | 1.51 | 4.29 | 1.56 | 4.12 | 1.54 | 4.15 | 1.54 | 4.29 | 1.56 |
| 10 | The NHS cares for us. | 5.2 | 1.48 | 4.64 | 1.58 | 5.14 | 1.52 | 5.17 | 1.48 | 5.07 | 1.51 | 5.29 | 1.48 |
| 11 | The NHS has gone out of its way to help people out. | 4.86 | 1.51 | 4.61 | 1.56 | 4.86 | 1.55 | 4.82 | 1.48 | 4.77 | 1.48 | 4.95 | 1.57 |
| | *Trust* | | | | | | | | | | | | |
| 12 | I trust my GP. | 4.95 | 1.51 | 4.48 | 1.57 | 4.94 | 1.54 | 4.88 | 1.5 | 4.82 | 1.55 | 5.05 | 1.44 |
| 13 | I trust the doctors and nurses working in NHS hospitals. | 5.27 | 1.48 | 4.71 | 1.49 | 5.22 | 1.53 | 5.23 | 1.45 | 5.14 | 1.51 | 5.37 | 1.43 |
| 14 | I trust the management that runs the NHS hospitals. | 3.33 | 1.57 | 3.51 | 1.59 | 3.41 | 1.58 | 3.29 | 1.57 | 3.29 | 1.55 | 3.43 | 1.6 |
| 15 | I trust the Department of Health and Social Care that manages the NHS. | 3.32 | 1.6 | 3.31 | 1.58 | 3.34 | 1.62 | 3.31 | 1.58 | 3.44 | 1.57 | 3.13 | 1.62 |
| | *Discrimination* | | | | | | | | | | | | |
| 16 | The quality of care the NHS delivers depends on where you live. | 4.87 | 1.51 | 5.03 | 1.39 | 4.78 | 1.59 | 4.98 | 1.43 | 4.77 | 1.52 | 5.08 | 1.46 |
| 17 | The quality of care the NHS delivers depends on your ethnic background. | 3.1 | 1.55 | 4.2 | 1.57 | 3.04 | 1.6 | 3.33 | 1.56 | 3.06 | 1.56 | 3.43 | 1.61 |
| 18 | The quality of care the NHS delivers depends on your country of origin. | 3.15 | 1.57 | 4.13 | 1.62 | 3.08 | 1.59 | 3.37 | 1.59 | 3.18 | 1.59 | 3.35 | 1.6 |

Note: Responses are measured on 1–7 Likert scales, where 1 = strongly disagree and 7 = strongly agree.

emerge in perceptions of ethical care, trust in healthcare professionals, and perceived discrimination. In these domains, ethnic minority respondents tend to report less positive evaluations of the NHS and stronger perceptions of inequity. These patterns are broadly consistent across both unadjusted and adjusted models, indicating that they are not fully explained by differences in age, gender, education, political orientation, or generalised trust.

**Table 3. Results of regression models assessing the relationship between ethnicity and trust in and satisfaction with the NHS in England.**

| Item | Statement | Unadjusted | | | Adjusted | | |
|---|---|---|---|---|---|---|---|
| | | b | Lower 95% CI | Upper 95% CI | b | Lower 95% CI | Upper 95% CI |
| | *Communication* | | | | | | |
| 1 | I believe the NHS is always honest in its communications. | −0.174 | −0.487 | 0.139 | −0.175 | −0.498 | 0.148 |
| 2 | The NHS is frank in its communications to people. | −0.148 | −0.452 | 0.155 | −0.134 | −0.448 | 0.179 |
| 3 | Information on important medical care issues is communicated openly by the NHS. | −0.393** | −0.689 | −0.097 | −0.447** | −0.753 | −0.141 |
| | *Competence* | | | | | | |
| 4 | The NHS is competent in providing a national healthcare service to the people of the UK. | −0.138 | −0.486 | 0.209 | −0.127 | −0.484 | 0.23 |
| 5 | The NHS knows how to supply the healthcare needs of our country. | −0.152 | −0.492 | 0.188 | −0.005 | −0.352 | 0.343 |
| 6 | I feel confident that the NHS can support and provide quality healthcare to the people in the UK. | 0 | −0.349 | 0.349 | −0.025 | −0.388 | 0.337 |
| | *Ethical care* | | | | | | |
| 7 | Promises made by the NHS are always delivered. | 0.023 | −0.269 | 0.315 | 0.042 | −0.262 | 0.347 |
| 8 | I can count on the NHS to do things in an ethical manner. | −0.689*** | −0.979 | −0.398 | −0.593*** | −0.891 | −0.295 |
| 9 | The NHS always put people's interests before its own. | −0.127 | −0.448 | 0.195 | −0.111 | −0.439 | 0.217 |
| 10 | The NHS cares for us. | −0.561*** | −0.871 | −0.252 | −0.529** | −0.844 | −0.214 |
| 11 | The NHS has gone out of its way to help people out. | −0.248 | −0.561 | 0.066 | −0.174 | −0.492 | 0.145 |
| | *Trust* | | | | | | |
| 12 | I trust my GP. | −0.466** | −0.779 | −0.153 | −0.415* | −0.738 | −0.093 |
| 13 | I trust the doctors and nurses working in NHS hospitals. | −0.56*** | −0.867 | −0.254 | −0.429** | −0.74 | −0.118 |
| 14 | I trust the management that runs the NHS hospitals. | 0.184 | −0.141 | 0.51 | 0.012 | −0.322 | 0.346 |
| 15 | I trust the Department of Health and Social Care that manages the NHS. | −0.019 | −0.349 | 0.312 | −0.097 | −0.442 | 0.247 |
| | *Discrimination* | | | | | | |
| 16 | The quality of care the NHS delivers depends on where you live. | 0.158 | −0.154 | 0.47 | 0.147 | −0.181 | 0.476 |
| 17 | The quality of care the NHS delivers depends on your ethnic background. | 1.103*** | 0.781 | 1.426 | 0.804*** | 0.471 | 1.136 |
| 18 | The quality of care the NHS delivers depends on your country of origin. | 0.981*** | 0.655 | 1.307 | 0.735*** | 0.395 | 1.076 |

Notes: Entries report linear regression coefficients (b) representing the mean difference in agreement scores between ethnic minority and White respondents (ethnic minority minus White). Outcomes are measured on 7-point Likert scales (1 = strongly disagree, 7 = strongly agree). Higher values indicate more positive evaluations of the NHS, except for items 16–18 (discrimination items), where higher values indicate stronger perceived inequity.

## Communication and competence

In the communication domain (statements 1–3), respondents were asked about the NHS's honesty, frankness, and openness in communication. We found no significant ethnic differences for statements concerning honesty ("I believe the

NHS is always honest in its communications") or frankness ("The NHS is frank in its communications"). By contrast, a significant ethnic difference emerged for perceptions of openness in communication ("Information on important medical care issues is communicated openly by the NHS."). In the unadjusted analysis, ethnic minority respondents reported lower agreement than White respondents (b = −0.39, 95% CI −0.69 to −0.10). This difference remained statistically significant after adjustment for demographic and attitudinal covariates (b = −0.45, 95% CI −0.75 to −0.14). These results indicate that while perceptions of honesty and frankness did not differ by ethnicity, ethnic minority respondents reported lower evaluations of the NHS's openness in communicating important healthcare information.

For the competence domain (statements 4–6), we found no statistically significant ethnic differences. Across all three items assessing perceptions of NHS competence and capability, the estimated differences between ethnic minority and White respondents were small and not statistically significant in either unadjusted or adjusted models. This included statements relating to the NHS's ability to provide a national healthcare service and confidence in its capacity to deliver quality care. Overall, perceptions of NHS competence did not vary by ethnicity.

### Ethical care and trust in providers

In contrast, ethnic differences were more evident in perceptions of NHS ethics and personal care, as well as in trust toward healthcare providers. Statements 7−11 addressed whether the NHS behaves ethically and demonstrates care toward patients. In two items (statements 8 and 10) ethnic minority respondents reported lower agreement than White respondents in both the unadjusted and the adjusted models. For example, on the statement "I can count on the NHS to do things in an ethical manner" (statement 8), ethnic minority participants reported significantly lower agreement than White respondents (adjusted b = −0.59, 95% CI −0.89 to −0.3).

Turning to trust in providers, we observed a mixed pattern across different levels of the health system. Trust in front-line providers showed clear ethnic differences. Ethnic minority respondents reported significantly lower trust in their personal GP than White respondents. For the statement "I trust my GP" (statement 12), the adjusted model indicated a substantial negative difference for ethnic minority respondents (b = −0.42, 95% CI −0.74 to −0.09). A similar pattern was observed for trust in hospital clinicians. On the statement "I trust the doctors and nurses working in NHS hospitals" (statement 13), ethnic minority respondents again reported lower agreement than White respondents (adjusted b = −0.43, 95% CI −0.74 to −0.12).

By contrast, trust in NHS leadership and management did not differ significantly by ethnicity. Statements 14 and 15 assessed trust in hospital management and iconsin the Department of Health and Social Care. For both items, the estimated differences between ethnic minority and White respondents were small and not statistically significant. Although the point estimates were slightly positive, indicating marginally higher reported trust among ethnic minority respondents, confidence intervals included zero, indicating no reliable ethnic differences. Overall, trust in NHS management and national health authorities appeared broadly similar across ethnic groups.

### Perceptions of discrimination in care

Statistically significant ethnic differences were most clearly evident for items assessing perceived structural unfairness or discrimination in healthcare (statements 16–18). These items asked whether respondents believed that the quality of NHS care depends on non-medical factors, specifically place of residence, ethnic background, or country of origin.

For the item concerning geographic variation in care (statement 16), the estimated ethnic difference was small and not statistically significant in either unadjusted or adjusted models. However, statistically significant ethnic differences emerged for the two items explicitly addressing perceived ethnic and national-origin discrimination in care. Ethnic minority respondents were significantly more likely than White respondents to agree that "The quality of care the NHS delivers depends on your ethnic background" (statement 17). In the adjusted model, the estimated ethnic difference was large and

positive (b = 0.8, 95% CI 0.47 to 1.14). A similarly pronounced pattern was observed for the statement "The quality of care the NHS delivers depends on your country of origin" (statement 18), where ethnic minority respondents again reported substantially higher agreement than White respondents (adjusted b = 0.74, 95% CI 0.4 to 1.08).

## Discussion

### Main finding of this study

This study documents clear ethnic differences in how people perceive and experience the NHS in England, with particularly pronounced gaps in perceptions of care, ethics, and discrimination. While ethnic minority and White respondents expressed broadly similar views regarding system competence and trust in NHS leadership, substantial differences emerged in perceptions of whether care quality depends on ethnic background or country of origin. Ethnic minority respondents were also more likely to report lower trust in front-line providers and to feel that the NHS does not consistently care for or support patients. Importantly, these differences were not uniform across all dimensions of healthcare experience. We found no ethnic gaps in perceptions of NHS honesty, frankness, promise-keeping, or overall competence. Instead, the largest disparities centred on relational and experiential aspects of care: feeling cared for, being treated ethically, and being protected from discrimination. Together, these findings suggest that ethnic inequalities in healthcare perceptions are less about assessments of technical capability and more about perceived fairness, responsiveness, and interpersonal treatment within the health system.

### What is already known on this topic

Previous research has consistently shown that ethnic minority patients, on average, report lower satisfaction with healthcare services than White patients [3,9,13]. Large-scale surveys in England, including the GP Patient Survey and other national audits, have documented an "ethnicity effect" in patient experience, even after adjusting for socioeconomic factors [10,13]. Prior studies have also reported lower levels of trust in the NHS and in healthcare providers among ethnic minority communities [15,16]. Existing research suggests that structural inequalities, past negative experiences, and broader social factors are associated with this trust gap [4–6,16]. However, much of this literature relies on aggregate measures of satisfaction or trust, or focuses on specific care settings (such as primary care or cancer services). As a result, there is limited evidence on which specific dimensions of healthcare experience contribute most to observed ethnic differences.

### What this study adds

This study extends existing research by identifying the specific dimensions of healthcare experience in which ethnic differences are most pronounced. Our findings show that ethnic minority patients in England do not express uniformly lower evaluations of the NHS. Instead, differences are concentrated in particular domains. Ethnic minority respondents reported significantly lower agreement with statements relating to ethical behaviour and personal care (such as whether the NHS behaves ethically or cares for patients) and reported lower levels of trust in front-line clinicians, including GPs and hospital doctors and nurses. In contrast, we found no ethnic differences of the NHS's overall competence or in trust toward NHS leadership and national health authorities.

Perhaps most importantly, this study demonstrates that perceptions of discrimination in NHS care vary sharply by ethnicity. Ethnic minority respondents were substantially more likely than White respondents to agree that the quality of NHS care depends on ethnic background and country of origin. These differences were among the largest observed across all survey items. While the study cannot determine whether such perceptions reflect individual experiences or broader societal narratives, their prevalence among ethnic minority respondents, and relative absence among White respondents, represents a significant empirical finding. It indicates a pronounced divergence in how different groups perceive fairness and equity within the health system.

By examining multiple dimensions of healthcare experience simultaneously, including communication, ethical treatment, perceived competence, and discrimination, this study provides a more granular picture of ethnic differences in trust and satisfaction with the NHS. The observed pattern, in which ethnic differences were larger for perceptions of care, fairness, and equal treatment than for assessments of technical competence, suggests that disparities in trust are more closely associated with relational and experiential aspects of care than with evaluations of clinical capability [27]. While causal mechanisms cannot be established with cross-sectional survey data, these findings indicate that efforts to reduce ethnic inequalities in patient trust may benefit from attention to interpersonal interactions and institutional practices, alongside technical performance. More broadly, the results highlight the multi-dimensional nature of ethnic inequalities in healthcare trust, consistent with existing calls to improve the inclusiveness and responsiveness of NHS services in a diverse population [6,17].

## Limitations of this study

We acknowledge several limitations. First, our measure of ethnicity was broad. We compared a combined "ethnic minority" group to "White" respondents, which obscures substantial heterogeneity within minority communities. The UK's ethnic minority population includes people of South Asian, Black African, Black Caribbean, East Asian, Middle Eastern, mixed and other backgrounds, whose experiences with the NHS may differ in important ways. Prior research suggests that some groups (for example, Pakistani or Bangladeshi patients) report particularly negative healthcare experiences, whereas others (such as some Indian patients) report experiences closer to the national average [12]. Owing to sample size constraints, we were unable to examine differences within minority subgroups. As a result, the estimates presented here represent average differences across diverse populations and may conceal important variation between specific ethnic groups. Future research with larger and more diverse samples would be well placed to disaggregate these categories and provide more targeted insights.

Second, the number of ethnic minority respondents in the survey was relatively small (n = 98). While this sample size was sufficient to detect medium-to-large differences between groups, smaller effects may not have been detected, and non-significant findings should therefore not be interpreted as evidence of no difference. In addition, ethnic minority respondents comprised approximately 8.82% of the sample, which is below their proportion in the English population. Under-representation of minority groups is a well-documented challenge in survey research [18], reflecting factors such as mistrust of research, language barriers, and survey design [24]. As a result, the minority respondents included here may not fully represent the experiences of all minority communities. Although we conducted robustness checks using post-stratification weights and found similar patterns of results, issues of representativeness remain. Accordingly, the findings should be interpreted with appropriate caution.

Third, the data are cross-sectional, capturing perceptions at a single point in time. Trust and satisfaction are dynamic and may change in response to new experiences or broader events (for example, high-profile developments within the NHS). As a result, we cannot infer causality or directionality from the observed associations. Negative experiences may reduce trust, but pre-existing mistrust could also shape how subsequent experiences are interpreted. Longitudinal research would be better suited to disentangling these dynamics. In addition, although we adjusted for relevant covariates (including age, education, political orientation and generalized trust), other potentially important factors (such as income, health status, or local service quality) were not measured and may also influence perceptions of care.

Finally, this study focused on subjective measures of patient experience and trust and did not include objective indicators of healthcare quality or clinical outcomes. While perceptions are themselves an important component of patient experience, future research could usefully link reported trust and satisfaction to observed healthcare encounters or outcomes (for example, examining whether lower trust in providers is associated with specific types of negative experiences or service use patterns). Such analyses were beyond the scope of the present study, but represent an important avenue for further investigation.

 

Despite these limitations, the findings have important implications for healthcare practice and policy. The observed perception gap between ethnic minority and White respondents highlights a potential disconnect in how different groups experience and evaluate NHS care. Even where overall levels of trust in the NHS remain relatively high, the results indicate that ethnic minority communities report systematically less positive perceptions in key areas of care and fairness. Addressing these differences is relevant not only for equity considerations but also for effective healthcare delivery, as prior research has shown that trust in providers and institutions is associated with care-seeking behaviour and adherence to medical advice [1].

## Supporting information

**S1 Table. Supplementary Table S1. Survey statements on attitudes and experiences of the NHS.** (DOCX)

**S2 Table. Supplementary Table S2. Demographic comparison of ethnic minority respondents with Census 2021 non-White population (England).** (DOCX)

**S3 Table. Supplementary Table S3. Full regression results for adjusted models.** (DOCX)

## Author contributions

**Conceptualization:** Steven David Pickering, Martin Ejnar Hansen, Han Dorussen, Jason Reifler, Thomas Scotto, Yosuke Sunahara, Dorothy Yen.

**Data curation:** Steven David Pickering.

**Formal analysis:** Steven David Pickering, Martin Ejnar Hansen.

**Funding acquisition:** Steven David Pickering, Han Dorussen, Jason Reifler, Thomas Scotto, Yosuke Sunahara, Dorothy Yen.

**Investigation:** Steven David Pickering.

**Methodology:** Steven David Pickering, Martin Ejnar Hansen.

**Project administration:** Steven David Pickering, Yosuke Sunahara.

**Writing – original draft:** Steven David Pickering, Martin Ejnar Hansen, Han Dorussen, Jason Reifler, Thomas Scotto, Yosuke Sunahara, Dorothy Yen.

**Writing – review & editing:** Steven David Pickering, Martin Ejnar Hansen, Han Dorussen, Jason Reifler, Thomas Scotto, Yosuke Sunahara, Dorothy Yen.

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
