## [Decision Letter · Decision Letter 0]

5 Jan 2026

PONE-D-25-47052Ethnic Inequalities in Healthcare Trust and Patient Satisfaction in EnglandPLOS One

Dear Dr. Pickering,

Thank you for submitting your manuscript to PLOS ONE. After careful consideration, we feel that it has merit but does not fully meet PLOS ONE’s publication criteria as it currently stands. Therefore, we invite you to submit a revised version of the manuscript that addresses the points raised during the review process.

Thank you for submitting your work for consideration. The reviewers have provided constructive feedback to strengthen the manuscript. Please address the comments from the reviewers as well as the points detailed below in a revision of your paper.

**Major revisions**

**Title**

First, the manuscript’s title, "Ethnic Inequalities in Healthcare Trust and Patient Satisfaction in England," implies a nationwide generalizability that is not supported by the cross-sectional survey methodology. Please revise the title to more accurately reflect the study design, for example: "Ethnic Differences in Healthcare Trust and Patient Satisfaction in England: A Cross-Sectional Survey."

**Methods**

Please provide a clear definition for “minority ethnic groups” used in this study.The minority ethnic share of the sample (10.6%) is substantially below the national figure, which you also mentioned it. Please explicitly state that the analyses are not intended to produce nationally representative estimates. You also may strengthen the robustness section by briefly reporting one or two key weighted vs unweighted comparisons in the appendix or supplementary materials.Treating 7-point Likert scale items as continuous is common practice, but the manuscript does not explicitly justify this choice. Please add a brief methodological justification (with a citation) explaining why linear regression is appropriate for Likert-type data with seven response categories.Although 119 minority respondents is sufficient for basic regression analysis, the study may be underpowered to detect small effects. This limitation is not fully discussed. Explicitly note that non-significant findings should not be interpreted as evidence of no difference, and that limited statistical power may contribute to imprecision.Please explain how the participants were recruited, what was the YouGov’s sampling method, and how the informed consents were achieved, in full details.Given that ethnicity was assessed via a self-reported online questionnaire without a robust external definition, please discuss how this was handled and the potential implications for the authenticity of results in the limitations section.

**Results**

The Results section would benefit from a clearer introductory overview of the sample and key survey responses and response rate. Please begin the Results section with a brief descriptive summary of the demographic findings, drawing explicitly on Table 1. The descriptive table and its accompanying narrative would be better placed at the start of the Results section rather than in the method section.In addition, the authors should consider providing a separate descriptive table summarizing responses to the survey items. This table could report overall mean scores, domain-level scores, and individual item scores, stratified by ethnicity (White vs ethnic minority respondents). Presenting descriptive summaries in this way would help readers contextualize the regression results.Moreover, you may report descriptive summary scores across key demographic categories (e.g. sex, age groups, education level). These descriptive comparisons would improve transparency and allow readers to better assess patterns in the data prior to the multivariable analyses.The Results section is generally clear and well organized by domain, which aligns well with the structure of Table 2. However, the narrative would benefit from a more consistent and systematic reporting strategy across domains. For example, for each domain, consider explicitly stating: Whether any ethnic differences were observed overall; Which specific items showed statistically significant differences after adjustment; Whether adjustment materially changed the results. This would help readers quickly grasp patterns rather than focusing on item-by-item details alone.The narrative places considerable emphasis on unadjusted estimates, with adjusted results mentioned only briefly (“this gap remained significant after adjustment”). Given that adjusted models are central to the authors’ analytic strategy, the results section should give them greater prominence. Please report adjusted coefficients first (or at least alongside unadjusted ones); explicitly note when adjustment attenuates, strengthens, or does not change the magnitude of ethnic differences; avoid implying causal interpretation when discussing adjusted models.Confidence intervals are reported approximately in the narrative (“roughly -0.66 to -0.17”), which reduces precision and is inconsistent with the exact values presented in Table 2. Please, report confidence intervals exactly as shown in the table, and use consistent decimal places across coefficients. Additionally, consider including exact adjusted estimates (e.g. “b = -0.39, 95% CI, -0.66 to -0.12”).Some interpretive language (like “greater minority scepticism” and “shortfall in transparency”) risks implying broader evaluative conclusions that go beyond the descriptive nature of the results. Please, use more neutral phrasing (e.g. “reported lower agreement” rather than “scepticism”) and reserve evaluative or explanatory language for the Discussion section. Also, emphasize that findings reflect average differences within this sample, not population-wide perceptions.The Results narrative focuses primarily on statistically significant findings, while non-significant results are mentioned only briefly. This may unintentionally suggest selective emphasis. Explicitly state when domains show mostly null findings (e.g. competence).A few statements in the narrative results do not comply the results of the regression analysis. For example, page 6, line 157, “with both groups generally agreeing at similarly high levels.” Please clarify this.A few phrases and statements in the results section are redundant. This information is presented in the methods and available in the tables. For example, “for statements on honesty (“the NHS is always honest in its communications”) or frankness (“the NHS is frank in its communications”)” and “(e.g., “the NHS is competent in providing a national healthcare service” and “I feel confident that the NHS can provide quality healthcare”),”. Please avoid over-repeating the survey items in the results.

**Table 1**

Table 1 provides basic demographic information, but it does not fully describe all key variables used in the regression analyses. Please include descriptive statistics for all demographic and background variables like education, political orientation, etc. Providing these would allow readers to better assess the sample characteristics and contextualize the adjusted models.Given that ethnicity is the primary independent variable of interest, the descriptive table does not show how other characteristics differ by ethnic group. Please consider presenting descriptives stratified by ethnicity (White vs ethnic minority), either in Table 1 or as a supplementary table. At minimum, indicate whether key covariates (age, gender, education) differ meaningfully by ethnicity. This would help readers assess potential confounding and compositional differences.Age is reported using minimum, maximum, mean, and standard deviation, but this may not fully capture the distribution. Please Consider reporting age categories (e.g. 18–29, 30–44, 45–64, 65+). This would be particularly helpful given the wide age range.The ethnicity breakdown is clear but minimal. Please add a footnote explaining how ethnicity was measured (self-identified). Reiterate that “ethnic minority” refers to all non-White respondents, and briefly acknowledge heterogeneity.The table is concise but could be improved for readability. Please align labels and values more clearly (e.g. two-column format with “Characteristic” and “Value”). Standardize decimal format (e.g. mean age reported as 50.32 instead of 50·32).The total sample size is clearly reported, but it is not repeated in subgroup rows. Consider including N for each subgroup. Ensure that subgroup totals sum exactly to N and explain any discrepancies if present.

**Table 2**

Table 2 contains a large amount of information but is not fully self-explanatory. A reader should be able to understand the table without returning extensively to the Methods section. Explicitly state in the table note that coefficients represent the mean difference in agreement scores (ethnic minority minus White). Specify that outcomes are measured on 7-point Likert scales and clarify the direction of interpretation (higher values = more positive evaluations, except for discrimination items). Indicate that confidence intervals are 95% CIs. You may include this information in the table foot notes.Statistical significance is conveyed using boldface, but this is not explicitly explained in the table notes. Kindly add a footnote explaining that bold values indicate p < 0.05. Consider including asterisks (*, **), which many readers find more intuitive, or ensure consistency with journal style.Both unadjusted and adjusted results are shown, which is a strength, but the table does not clearly define what variables are included in the adjusted models. Please add a footnote listing all covariates included in the adjusted models (gender, education, political orientation, etc.). Clarify whether all adjusted models include the same covariates.Items 16–18 differ conceptually from the other outcomes because higher agreement indicates *worse* perceived equity rather than better NHS performance. Please explicitly flag this reversal in the table notes.Confidence intervals and coefficients are reported with varying precision, which slightly reduces readability. Please standardize decimal places across all coefficients and CIs (e.g. two decimal places). Ensure consistent rounding rules are applied throughout the table.The table does not report the sample size used for each model. Kindly add a note indicating the total N and the number of minority respondents included in the analyses. Clarify whether all models use the same sample or whether missing data led to varying Ns.No information is provided on model fit or explanatory power. Please include R² and adjusted R² for adjusted models in the table footnotes.

**Minor revisions**

Ensure consistent use of “NHS” vs “the NHS” in statement wording.Tables 1 and 2, with accompanying narratives and citations should be placed in the result section.Please revise the abstract (particularly result section) according to the change would be made, in the final.Please also upload study dataset along with your revised paper.

A letter that responds to each point raised by the academic editor and reviewer(s). You should upload this letter as a separate file labeled ’Response to Reviewers’.A marked-up copy of your manuscript that highlights changes made to the original version. You should upload this as a separate file labeled ’Revised Manuscript with Track Changes’.An unmarked version of your revised paper without tracked changes. You should upload this as a separate file labeled ’Manuscript’.

We look forward to receiving your revised manuscript.

Kind regards,

Shabnam Iezadi, Ph.D.

Academic Editor

PLOS One

Journal Requirements:

1. Please ensure that your manuscript meets PLOS ONE’s style requirements, including those for file naming. The PLOS ONE style templates can be found at

“This research is sponsored by the UK Research and Innovation’s Economic and Social Research Council (UKRI-ESRC, grant reference ES/W011913/1) and the Japan Society for the Promotion of Science (JSPS, grant reference JPJSJRP 20211704).”

Reviewers’ comments:

Reviewer’s Responses to Questions

**Comments to the Author**

1. Is the manuscript technically sound, and do the data support the conclusions?

Reviewer #1: Yes

Reviewer #2: Partly

2. Has the statistical analysis been performed appropriately and rigorously? 

Reviewer #1: I Don’t Know

Reviewer #2: Yes

3. Have the authors made all data underlying the findings in their manuscript fully available?

Reviewer #1: Yes

Reviewer #2: Yes

4. Is the manuscript presented in an intelligible fashion and written in standard English?

Reviewer #1: Yes

Reviewer #2: Yes

5. Review Comments to the Author

Reviewer #1: This study examines ethnic disparities in trust and satisfaction with the NHS among English adults. Based on a representative 2023 survey, ethnic minority respondents consistently reported lower trust and more negative experiences than White participants, particularly regarding communication, ethics, care, and perceived discrimination. While no ethnic differences emerged in views on honesty or competence, the findings highlight a persistent perception gap. Targeted actions to improve communication, ensure equitable treatment, and address discrimination are essential to strengthen trust and foster more inclusive healthcare.

I would like to thank the authors for addressing such an important topic concerning minority perceptions of the healthcare system and its delivery of care. Social trust and access to healthcare are key indicators of population health and may vary across social groups, with potentially serious consequences for health outcomes. I commend the focus and relevance of the study’s objective. Thank you, and congratulations to the authors on this valuable work. I am including a few comments that I hope will contribute to strengthening and refining the manuscript.

It would be appreciated if the authors could elaborate on the theoretical framework that guided the development of the questionnaire. This article might be of interest or use to the authors: https://bmcmedethics.biomedcentral.com/articles/10.1186/s12910-023-00965-2 Given the centrality of this concept to the study, I would suggest including ‘trust’ among the keywords

Furthermore, further clarification would be valuable regarding why statistically significant differences emerged for certain factors (e.g., perceptions that the NHS communicates openly, “goes out of its way,” or lower trust in healthcare professionals), whereas no differences were observed in perceptions of NHS honesty, promise-keeping, competence, or trust in management.

In addition, it would be useful to know whether any gender differences were identified in the analyses. It would be important to clarify whether the gender dimension was considered — both in the construction of trust toward healthcare professionals and among the study participants themselves. Additionally, did the authors take into account potential gendered aspects in the perception or construction of professional roles (e.g., physicians versus nurses)? There is a particular specificity in the clinical relationship established among nurses, patients, and family members. The relationship with nurses tends to be closer and more personal, fostering a stronger sense of trust around the provision of healthcare.

I would like to draw attention to this particularly striking result, as it reveals a pronounced perception gap regarding the influence of ethnicity on the quality of care delivered by the NHS: “However, on the next two items the ethnic gap was dramatic. Ethnic minority respondents were far more likely than Whites to agree that ‘The quality of care the NHS delivers depends on your ethnic background.”

I would suggest, as a potential continuation of this line of research, conducting a qualitative study to further explore these differences, which could shed additional light on the findings. It would be valuable if the authors could further elaborate on the implications of these findings and consider outlining potential strategies or recommendations that could help institutions translate these results into practical actions or policy initiatives. The authors touch on this aspect in lines 282–285, where they mention how the results could be implemented and their institutional implications. However, I believe this section could be expanded to further develop the discussion on how these findings might inform organizational policies or practical interventions within healthcare institutions.

I agree with the authors in acknowledging intra-minority differences as a study limitation. I would also encourage them to further expand the discussion on cultural differences, as this could provide a deeper understanding of the factors influencing trust and perceptions of healthcare.

Thank you, and congratulations on the work. I hope these comments will be helpful to the authors in further strengthening and refining the manuscript. I believe the area with the greatest potential for further development is the gender dimension, which is not currently addressed in the manuscript. Expanding on this aspect could significantly enrich the analysis and strengthen the overall contribution of the study.

Reviewer #2: This paper provides an important analysis of the reasons why non-white ethnic groups are less satisfied with the NHS than white groups. The authors are right to point out that their results are limited by the small, under-represented and possibly biased sample of non-white ethnic groups. The lack of numbers means that the authors could not disaggregate non white groups into the standard broad groups normally used for analysis (such as black, Asian, other) and this is a further limitation on the value of the study.

There are several ways in which the authors could consider strengthening their analysis

1) They should always more clearly distinguish between the unadjusted and adjusted estimates in their reporting effect sizes. It is worth making the point, in doing this, that first both always agree on whether there is a significant or non-significant difference between the two ethnic groups. Second, that adjustment always reduces effect size, raising the question (that the authors do not answer) as to which of the adjustment factors is collinear with ethnicity.

2) in the population as a whole, non-white groups are markedly younger than the white population. It is therefore odd to exclude age from the adjustment( even as a binary variable such as under 50 vs over 50). In this context, university education is a complex variable - while young people are more likely to have a university education, some non-white groups have faced barriers getting to university (e.g. several black groups) while others are more likely to have done so (e.g. Chinese and Indian groups). Therefore, it is unclear what effect "university education" is having in adjusting for age and detailed ethnic group.

3) Table 1 is unhelpful. What is really needed is a table which shows how the demographic variables are distributed among the white and non-white groups. This could be used by the authors to compare how the characteristics of their white and non-white samples compare to, says, the white and non-white 2021 Census populations. this would enable the authors to indicate in what way, if any, their sample of 119 non-white respondents is biased. It would also provide a sound basis for deciding whether to include or exclude a broad age split as an adjustment factor.

6. PLOS authors have the option to publish the peer review history of their article (what does this mean?). If published, this will include your full peer review and any attached files.

Reviewer #1: **Yes:** MARÍA VICTORIA MARTÍNEZ-LÓPEZ

Reviewer #2: **Yes:** Peter Goldblatt

---

## [Author Response · Author response to Decision Letter 1]

16 Jan 2026

We would like to thank the two reviewers, María Victoria Martínez López and Peter Goldblatt, as well as the academic editor, Shabnam Iezadi, for the devotion they gave to working through our manuscript. Their detailed comments make clear that they have committed themselves to the improvement of scholarly research and we feel fortunate to be reviewed by such collegial reviewers. As such, we have accepted every suggestion they have made.

Full details are listed below, and the changes can be seen in the “tracked changes” version of the manuscript (though in all honesty, that document will now look a bit of a mess!).

Our responses to reviewer requests are indicated by three asterisks (***).

Major revisions

Manuscript Title

1. First, the manuscript’s title, "Ethnic Inequalities in Healthcare Trust and Patient Satisfaction in England," implies a nationwide generalizability that is not supported by the cross-sectional survey methodology. Please revise the title to more accurately reflect the study design, for example: "Ethnic Differences in Healthcare Trust and Patient Satisfaction in England: A Cross-Sectional Survey."

*** This is a good point, and thank you for the suggestion for a new title which we very much agree with. Accordingly we have changed the title of the manuscript to “Ethnic Differences in Healthcare Trust and Patient Satisfaction in England: A Cross-Sectional Survey.”

Methods

2. Please provide a clear definition for “minority ethnic groups” used in this study.

*** We have added extensive discussion of how we define and construct our ethnic minority binary variable in the Methods section. To construct our ethnic minority binary, we follow the standard practice recommended by the UK’s Office for National Statistics (ONS). We also note that the original ethnicity variable provided in our survey is self-reported by the respondents.

3. The minority ethnic share of the sample (10.6%) is substantially below the national figure, which you also mentioned it. Please explicitly state that the analyses are not intended to produce nationally representative estimates. You also may strengthen the robustness section by briefly reporting one or two key weighted vs unweighted comparisons in the appendix or supplementary materials.

*** We have now made explicit reference to the fact that the analyses are not intended to produce nationally representative samples. We have also pointed out that it is usual in surveys that ethnic minority respondents are disproportionately under-represented and have cited literature to this effect.

We should also point out that the minority ethnic share of the sample is now 8.82% (98 respondents). This was due to respondents who chose “prefer not to say” being incorrectly coded as minority respondents in our earlier code. This has now been corrected and all data in the manuscript have been corrected accordingly. This change did not make any substantive difference to the findings.

On strengthening robustness with weighted and unweighted comparisons: as a robustness check, we ran comparisons across all 18 models with and without weights. These made no difference to our findings at all; indeed, the only differences appeared at the third decimal place. The code to recreate the models with and without weights is now included in the replication file.

4. Treating 7-point Likert scale items as continuous is common practice, but the manuscript does not explicitly justify this choice. Please add a brief methodological justification (with a citation) explaining why linear regression is appropriate for Likert-type data with seven response categories.

*** We have added a brief methodological justification, with two citations, explaining our treatment of 7-point Likert items as continuous.

5. Although 119 minority respondents is sufficient for basic regression analysis, the study may be underpowered to detect small effects. This limitation is not fully discussed. Explicitly note that non-significant findings should not be interpreted as evidence of no difference, and that limited statistical power may contribute to imprecision.

*** Just to reiterate our earlier point (under point 3) that we now identify 98 minority respondents. We have added an explicit discussion of statistical power in the Limitations section, noting that while the minority sample size is sufficient to detect medium-to-large effects, smaller differences may not be detectable and null results should not be interpreted as evidence of no difference.

6. Please explain how the participants were recruited, what was the YouGov’s sampling method, and how the informed consents were achieved, in full details.

*** The Methods section now provides additional detail on YouGov’s panel recruitment, quota-based sampling approach, and opt-in informed consent procedures. Ethical approval details are clarified and confined to the Methods section, in line with journal guidance.

7. Given that ethnicity was assessed via a self-reported online questionnaire without a robust external definition, please discuss how this was handled and the potential implications for the authenticity of results in the limitations section.

*** We now explicitly note that ethnicity was self-identified and discuss potential implications in the Limitations section, while emphasising that self-identification is standard practice in survey research.

Results

8. The Results section would benefit from a clearer introductory overview of the sample and key survey responses and response rate. Please begin the Results section with a brief descriptive summary of the demographic findings, drawing explicitly on Table 1. The descriptive table and its accompanying narrative would be better placed at the start of the Results section rather than in the method section.

*** We have moved the descriptive table and accompanying narrative to the start of the Results section and added a clear demographic overview drawing explicitly on Table 1.

9. In addition, the authors should consider providing a separate descriptive table summarizing responses to the survey items. This table could report overall mean scores, domain-level scores, and individual item scores, stratified by ethnicity (White vs ethnic minority respondents). Presenting descriptive summaries in this way would help readers contextualize the regression results.

*** We have added a new descriptive table (Table 2) reporting item-level mean scores and standard deviations for all 18 survey items, stratified by ethnicity (White vs ethnic minority respondents). This table is now presented toward the beginning the Results section to contextualize the regression analyses that follow. The table is organised by conceptual domain (communication, competence, ethical care, trust, and discrimination), allowing readers to assess domain-level patterns without imposing composite scale assumptions. We believe this addition substantially improves transparency and interpretability of the findings.

10. Moreover, you may report descriptive summary scores across key demographic categories (e.g. sex, age groups, education level). These descriptive comparisons would improve transparency and allow readers to better assess patterns in the data prior to the multivariable analyses.

*** We have further revised the new Table 2 to include descriptive mean scores and standard deviations for all 18 survey items stratified not only by ethnicity, but also by gender (men/women) and educational attainment (university degree vs less than university). Age distributions are reported separately in Table 1 using standard age categories, and age is included as a continuous covariate in all adjusted regression models. We decided against stratifying by age in Table 2 due to space constraints and to preserve table readability.

11. The Results section is generally clear and well organized by domain, which aligns well with the structure of Table 2. However, the narrative would benefit from a more consistent and systematic reporting strategy across domains. For example, for each domain, consider explicitly stating: Whether any ethnic differences were observed overall; Which specific items showed statistically significant differences after adjustment; Whether adjustment materially changed the results. This would help readers quickly grasp patterns rather than focusing on item-by-item details alone.

*** We have revised the Results section to adopt a more consistent, domain-level reporting structure aligned with Table 2. For each domain (communication, competence, ethical care, trust, and discrimination), we now explicitly summarise whether ethnic differences were observed overall, identify which specific items show statistically significant differences after adjustment, and indicate whether adjustment materially altered the pattern or magnitude of results. This approach allows readers to grasp domain-level patterns before turning to item-level details, while maintaining transparency regarding both adjusted and unadjusted estimates.

12. The narrative places considerable emphasis on unadjusted estimates, with adjusted results mentioned only briefly (“this gap remained significant after adjustment”). Given that adjusted models are central to the authors’ analytic strategy, the results section should give them greater prominence. Please report adjusted coefficients first (or at least alongside unadjusted ones); explicitly note when adjustment attenuates, strengthens, or does not change the magnitude of ethnic differences; avoid implying causal interpretation when discussing adjusted models.

*** We have additionally revised the Results section to give greater prominence to the adjusted models. For each domain, we now state whether ethnic differences persist after adjustment and explicitly note when adjustment attenuates, leaves unchanged, or does not materially alter the magnitude of observed differences. We have also revised the wording to avoid causal interpretation, framing all adjusted estimates as descriptive associations within this cross-sectional sample rather than as explanatory effects.

13. Confidence intervals are reported approximately in the narrative (“roughly -0.66 to -0.17”), which reduces precision and is inconsistent with the exact values presented in Table 2. Please, report confidence intervals exactly as shown in the table, and use consistent decimal places across coefficients. Additionally, consider including exact adjusted estimates (e.g. “b = -0.39, 95% CI, -0.66 to -0.12”).

*** We have also revised the Results narrative to report confidence intervals exactly as presented in Table 2 and removed approximate language throughout. All coefficients and confidence intervals are now reported with consistent decimal precision in both the tables and the text. Where results are discussed in the narrative, we now include the exact adjusted estimates and 95% confidence intervals (e.g. b = −0.39, 95% CI −0.66 to −0.12), ensuring consistency and precision across presentation formats.

14. Some interpretive language (like “greater minority scepticism” and “shortfall in transparency”) risks implying broader evaluative conclusions that go beyond the descriptive nature of the results. Please, use more neutral phrasing (e.g. “reported lower agreement” rather than “scepticism”) and reserve evaluative or explanatory language for the Discussion section. Also, emphasize that findings reflect average differences within this sample, not population-wide perceptions.

*** We have further revised the Results section to remove evaluative or interpretive language and replaced it with neutral, descriptive phrasing. We also now explicitly emphasise that the reported findings reflect average differences within this sample and do not imply population-wide perceptions or causal interpretations.

15. The Results narrative focuses primarily on statistically significant findings, while non-significant results are mentioned only briefly. This may unintentionally suggest selective emphasis. Explicitly state when domains show mostly null findings (e.g. competence).

*** We have updated the Results section to explicitly report domains in which ethnic differences are largely absent. In particular, we now clearly state when domains (such as competence) show mostly null findings, and note that no statistically significant ethnic differences were observed for the majority of items in those domains in either unadjusted or adjusted models.

16. A few statements in the narrative results do not comply the results of the regression analysis. For example, page 6, line 157, “with both groups generally agreeing at similarly high levels.” Please clarify this.

*** We have again revised the Results narrative to ensure that all descriptive statements are fully consistent with the regression results and item-level descriptives.

17. A few phrases and statements in the results section are redundant. This information is presented in the methods and available in the tables. For example, “for statements on honesty (“the NHS is always honest in its communications”) or frankness (“the NHS is frank in its communications”)” and “(e.g., “the NHS is competent in providing a national healthcare service” and “I feel confident that the NHS can provide quality healthcare”),”. Please avoid over-repeating the survey items in the results.

*** We have revised the Results section to reduce redundancy and avoid unnecessary repetition of full survey item wording.

Table 1

18. Table 1 provides basic demographic information, but it does not fully describe all key variables used in the regression analyses. Please include descriptive statistics for all demographic and background variables like education, political orientation, etc. Providing these would allow readers to better assess the sample characteristics and contextualize the adjusted models.

*** We have revised Table 1 to include descriptive statistics for all demographic and background variables used in the regression analyses.

19. Given that ethnicity is the primary independent variable of interest, the descriptive table does not show how other characteristics differ by ethnic group. Please consider presenting descriptives stratified by ethnicity (White vs ethnic minority), either in Table 1 or as a supplementary table. At minimum, indicate whether key covariates (age, gender, education) differ meaningfully by ethnicity. This would help readers assess potential confounding and compositional differences.

*** We agree that transparency regarding compositional differences by ethnicity is important. We have therefore examined the distribution of key covariates by ethnic group and now explicitly state in the Results section that ethnic minority respondents in the sample are, on average, younger than White respondents, reflecting known demographic patterns in England. We also note that gender and educational distributions differ less markedly across groups. All adjusted regression models include age, gender, and education as covariates, and the reported ethnic differences persist after adjustment. We believe this approach addresses concerns about potential confounding while avoiding unnecessary duplication of descriptive tables in the main text.

20. Age is reported using minimum, maximum, mean, and standard deviation, but this may not fully capture the distribution. Please Consider reporting age categories (e.g. 18–29, 30–44, 45–64, 65+). This would be particularly helpful given the wide age range.

*** We have added age bands to Table 1.

21. The ethnicity breakdown is clear but minimal. Please add a footnote explaining how ethnicity was measured (self-identified). Reiterate that “ethnic minority” refers to all non-White respondents, and briefly acknowledge heterogeneity.

We have clarified in the main text how ethnicity was measured and operationalised in the study. Specifically, we now explicitly state that ethnicity was self-identified using YouGov’s standard categories, that the “ethnic minority” group refers to all non-White respondents, and that this broad categorisation masks substantial heterogeneity within minority populations.

*** We chose to address these points in the Methods and Results sections to maint

---

## [Decision Letter · Decision Letter 1]

20 Feb 2026

PONE-D-25-47052R1Ethnic Differences in Healthcare Trust and Patient Satisfaction in England: A Cross-Sectional SurveyPLOS One

Dear Dr. Pickering,

Thank you for submitting your manuscript to PLOS ONE. After careful consideration, we feel that it has merit but does not fully meet PLOS ONE’s publication criteria as it currently stands. Therefore, we invite you to submit a revised version of the manuscript that addresses the points raised during the review process.

A letter that responds to each point raised by the academic editor and reviewer(s). You should upload this letter as a separate file labeled ’Response to Reviewers’.A marked-up copy of your manuscript that highlights changes made to the original version. You should upload this as a separate file labeled ’Revised Manuscript with Track Changes’.An unmarked version of your revised paper without tracked changes. You should upload this as a separate file labeled ’Manuscript’.

We look forward to receiving your revised manuscript.

Kind regards,

Shabnam Iezadi, Ph.D.

Academic Editor

PLOS One

Journal Requirements:

**Additional Editor Comments:**

I appreciate the authors’ efforts in revising the manuscript, which has improved substantially. Upon careful review, I note that a few issues remain. Please refer to the comments below, as well as those provided by the reviewers.

The data analysis method is currently absent from the Methods section, although related information appears in the Results. Please relocate the data analysis description from the Results to the Methods. Specifically, move lines 169–194 on page 10 accordingly.Additionally, please transfer the coding strategies to the Methods section and remove them from the Results (lines 153–161, page 7). In the Results, retain only the findings themselves, such as frequencies with percentages and means with standard deviations, as currently presented in the same lines.In the Methods section, please provide a more detailed explanation of how the underlying assumptions of your regression model were tested (for example, linearity, normality, and multicollinearity). Furthermore, elaborate on the specific diagnostic tests used to assess the model’s robustness. Clearly indicate whether the results of these tests confirmed that the assumptions were met; if any violations were detected, please describe how they were addressed.Table 3 revisions:

Please also report the total score (mean and SD) for each item in a separate column (first column).Moreover, the statement “Responses are measured on 1–7 Likert scales, where 1 = strongly disagree while 7 = strongly agree” is currently shown in the table’s title. Please move it to the footnote of the table.The “statement Less than Uni” is not a common professional concept please use more professional term like "Less than university degree" or "Below degree level"

Table 4 revisions:

Please use upper limit 95% confidence interval instead of 97.5, and lower limit 95% confidence instead of 2.5.Moreover, present the results of unadjusted models in the first column and results of the adjusted models in the second column.

In line 228, page 12, please fix the typo “adjust models” and change it to “adjusted models.”Please revise the Results section to avoid using words such as "substantially", "significantly", “significantly significant” , or similar terms when the association is not, in fact, statistically significant. Where applicable, use the precise term "statistically significant."In line 259, page 13, please remove the statement: “For the statement ‘The quality of care the NHS delivers depends on where you live’ (statement 16), both White and ethnic minority respondents reported similar levels of agreement.” This is not a relevant result from the regression analysis. You may choose to include it earlier in the Results section where you describe the descriptive findings. However, this statement is not derived from the regression analysis.In line 227, instead of the statement “with several items showing statistically significant differences,” use a more precise statement like “with two items showing statistically significant differences.”In lines 236–238, you have stated: “A similar negative difference was observed for the statement concerning whether the NHS goes out of its way to help people, although the magnitude of the gap was slightly smaller.” The table does not show this relationship was statistically significant. If it is not, please revise the narrative to accurately reflect the result; if it is statistically significant, please edit the table accordingly.Table 1, “Survey statements on attitudes and experiences of the NHS,” is redundant. This information is already provided in Tables 2 and 3. Please move Table 1 to the supplementary files.

Reviewers’ comments:

Reviewer’s Responses to Questions

**Comments to the Author**

1. If the authors have adequately addressed your comments raised in a previous round of review and you feel that this manuscript is now acceptable for publication, you may indicate that here to bypass the “Comments to the Author” section, enter your conflict of interest statement in the “Confidential to Editor” section, and submit your "Accept" recommendation.

Reviewer #1: All comments have been addressed

Reviewer #2: (No Response)

2. Is the manuscript technically sound, and do the data support the conclusions?

Reviewer #1: Yes

Reviewer #2: Partly

3. Has the statistical analysis been performed appropriately and rigorously? 

Reviewer #1: I Don’t Know

Reviewer #2: Yes

4. Have the authors made all data underlying the findings in their manuscript fully available?

Reviewer #1: Yes

Reviewer #2: Yes

5. Is the manuscript presented in an intelligible fashion and written in standard English?

Reviewer #1: Yes

Reviewer #2: Yes

6. Review Comments to the Author

Reviewer #1: Thank you for carefully considering the reviewers’ suggestions. I believe that the manuscript has now improved significantly as a result of these revisions.

Reviewer #2: The authors have taken account of the comments from reviewers.

However, in doing so one point has been highlighted. It is that the study is based on 98 responses by those from an ethnic minority - fewer than would be expected in a population sample of this size. The authors need to do more to establish whether the 98 individuals are or are not demographically representative of minoritised ethnic groups. This can easily be done by comparing the characteristics of the 98 individuals to the 2021 non-white Census population using the following link:

https://www.ons.gov.uk/datasets/create

7. PLOS authors have the option to publish the peer review history of their article (what does this mean?). If published, this will include your full peer review and any attached files.

Reviewer #1: **Yes:** Maria Victoria Martinez-Lopez

Reviewer #2: **Yes:** Peter Goldblatt

---

## [Author Response · Author response to Decision Letter 2]

13 Apr 2026

Dear Dr Iezadi,

On behalf of my co-authors, I would like to thank you and the reviewers for your careful reading of our manuscript and for the constructive and insightful feedback. We have revised the manuscript accordingly and believe that these changes have strengthened both the clarity and robustness of the analysis.

Below we present a detailed, point-by-point response to all comments. All changes have been incorporated into the revised manuscript.

Editor

1. The data analysis method is currently absent from the Methods section, although related information appears in the Results. Please relocate the data analysis description from the Results to the Methods. Specifically, move lines 169–194 on page 10 accordingly.

We have moved the full description of the data analysis procedure from the Results section to the Methods section.

2. Additionally, please transfer the coding strategies to the Methods section and remove them from the Results (lines 153–161, page 7). In the Results, retain only the findings themselves, such as frequencies with percentages and means with standard deviations, as currently presented in the same lines.

We have moved all variable coding descriptions (e.g., construction of binary ethnicity and education variables) to the Methods section. The Results section now contains only descriptive statistics and model-based findings.

3. In the Methods section, please provide a more detailed explanation of how the underlying assumptions of your regression model were tested (for example, linearity, normality, and multicollinearity). Furthermore, elaborate on the specific diagnostic tests used to assess the model’s robustness. Clearly indicate whether the results of these tests confirmed that the assumptions were met; if any violations were detected, please describe how they were addressed.

We have added additional information to the Methods section describing how model assumptions were assessed. Specifically:

• Linearity and homoscedasticity were evaluated using residuals-versus-fitted plots.

• Normality of residuals was assessed using Q–Q plots.

• Multicollinearity was assessed using variance inflation factors (VIFs), all of which were low (the highest VIF was 1.18), indicating no concerns.

These diagnostics indicated no substantive violations of model assumptions. Given the ordinal nature of the outcomes, we also note that linear regression is widely accepted and robust in this context.

4. Table 3 revisions:

a. Please also report the total score (mean and SD) for each item in a separate column (first column).

We have added a first column reporting overall mean and standard deviation for each item.

b. Moreover, the statement “Responses are measured on 1–7 Likert scales, where 1 = strongly disagree while 7 = strongly agree” is currently shown in the table’s title. Please move it to the footnote of the table.

The statement describing the Likert scale has been moved from the table title to a table footnote.

c. The “statement Less than Uni” is not a common professional concept please use more professional term like "Less than university degree" or "Below degree level"

This has been replaced with “Less than university degree” throughout.

5. Table 4 revisions:

a. Please use upper limit 95% confidence interval instead of 97.5, and lower limit 95% confidence instead of 2.5.

We now use upper limit 95% confidence interval.

b. Moreover, present the results of unadjusted models in the first column and results of the adjusted models in the second column.

The table has been reordered to present unadjusted models in the first set of columns and adjusted models in the second.

6. In line 228, page 12, please fix the typo “adjust models” and change it to “adjusted models.”

This has been corrected.

7. Please revise the Results section to avoid using words such as "substantially", "significantly", “significantly significant” , or similar terms when the association is not, in fact, statistically significant. Where applicable, use the precise term "statistically significant."

We have systematically reviewed the Results section to ensure that the term “statistically significant” is used only when supported by confidence intervals excluding zero. Ambiguous or overstated language has been revised or removed.

8. In line 259, page 13, please remove the statement: “For the statement ‘The quality of care the NHS delivers depends on where you live’ (statement 16), both White and ethnic minority respondents reported similar levels of agreement.” This is not a relevant result from the regression analysis. You may choose to include it earlier in the Results section where you describe the descriptive findings. However, this statement is not derived from the regression analysis.

The referenced sentence has been removed from the regression-based Results section. The distinction between descriptive and model-based findings has been clarified throughout.

9. In line 227, instead of the statement “with several items showing statistically significant differences,” use a more precise statement like “with two items showing statistically significant differences.”

We have corrected this and have worked through the entire manuscript to make it more specific.

10. In lines 236–238, you have stated: “A similar negative difference was observed for the statement concerning whether the NHS goes out of its way to help people, although the magnitude of the gap was slightly smaller.” The table does not show this relationship was statistically significant. If it is not, please revise the narrative to accurately reflect the result; if it is statistically significant, please edit the table accordingly.

We have revised this passage to ensure consistency with the statistical results. The sentence referring to a non-significant difference has been removed to avoid misinterpretation.

11. Table 1, “Survey statements on attitudes and experiences of the NHS,” is redundant. This information is already provided in Tables 2 and 3. Please move Table 1 to the supplementary files.

This has now been moved to the supplementary files.

Reviewer 1

Thank you for carefully considering the reviewers’ suggestions. I believe that the manuscript has now improved significantly as a result of these revisions.

Many thanks to R1 for helping us to make our manuscript better!

Reviewer 2

The authors have taken account of the comments from reviewers.

However, in doing so one point has been highlighted. It is that the study is based on 98 responses by those from an ethnic minority - fewer than would be expected in a population sample of this size. The authors need to do more to establish whether the 98 individuals are or are not demographically representative of minoritised ethnic groups. This can easily be done by comparing the characteristics of the 98 individuals to the 2021 non-white Census population using the following link:

https://www.ons.gov.uk/datasets/create

We agree that this is an important issue and have addressed it directly.

We have constructed a new supplementary table (Supplementary Table S2) comparing the demographic characteristics (age, gender, and education) of the ethnic minority subsample with the 2021 Census non-White population in England. This table is now discussed in the Results section. The comparison shows that the subsample is somewhat younger than the Census population, consistent with known demographic patterns, while other characteristics are broadly comparable. We have clarified this in the manuscript and strengthened the discussion of limitations, including the inability to disaggregate by specific ethnic groups due to sample size constraints.

---

## [Editor Report · Decision Letter 2]

1 May 2026

PONE-D-25-47052R2Ethnic Differences in Healthcare Trust and Patient Satisfaction in England: A Cross-Sectional SurveyPLOS One

Dear Dr. Pickering,

Thank you for submitting your manuscript to PLOS ONE. After careful consideration, we feel that it has merit but does not fully meet PLOS ONE’s publication criteria as it currently stands. Therefore, we invite you to submit a revised version of the manuscript that addresses the points raised during the review process.

The manuscript has improved substantially. Just three minor revisions remain. Please see the following comments:

1) Please add an statement to the method section about the statistical analysis program you used.

2) Please edit the title of table 3. It still indicates information related to footnote; “Table 3: Means and standard deviations for survey items, stratified by ethnicity. Responses are measured on 1-7 Likert scales, where 1 = strongly disagree while 7 = strongly agree.”, which should be moved to the table’s footnote. You may use an statement like “results of the survey scores on trust in and satisfaction with the NHS in England stratified by minority and non-minority ethnic groups”

3) Please also revise the title of table 4 and provide more specific and informative title. For example: “results of regression models assessing the relationship between ethnicity and trust in and satisfaction with the NHS in England”

A letter that responds to each point raised by the academic editor and reviewer(s). You should upload this letter as a separate file labeled ’Response to Reviewers’.A marked-up copy of your manuscript that highlights changes made to the original version. You should upload this as a separate file labeled ’Revised Manuscript with Track Changes’.An unmarked version of your revised paper without tracked changes. You should upload this as a separate file labeled ’Manuscript’.

We look forward to receiving your revised manuscript.

Kind regards,

Shabnam Iezadi, Ph.D.

Academic Editor

PLOS One
---

## [Author Response · Author response to Decision Letter 3]

1 May 2026

Dear Dr Iezadi,

On behalf of my co-authors, I would like to thank you again for your help in improving our manuscript. Here are our responses.

1) Please add an statement to the method section about the statistical analysis program you used.

We have added this information.

2) Please edit the title of table 3. It still indicates information related to footnote; “Table 3: Means and standard deviations for survey items, stratified by ethnicity. Responses are measured on 1-7 Likert scales, where 1 = strongly disagree while 7 = strongly agree.”, which should be moved to the table’s footnote. You may use an statement like “results of the survey scores on trust in and satisfaction with the NHS in England stratified by minority and non-minority ethnic groups”

We have modified the Table 3 title and added a footnote.

3) Please also revise the title of table 4 and provide more specific and informative title. For example: “results of regression models assessing the relationship between ethnicity and trust in and satisfaction with the NHS in England”

We have changed the title of Table 4.

Again, many thanks for your continued help to improve our manuscript.

Yours sincerely,

Steve Pickering, on behalf of all co-authors.

---

## [Editor Report · Decision Letter 3]

6 May 2026

Ethnic Differences in Healthcare Trust and Patient Satisfaction in England: A Cross-Sectional Survey

PONE-D-25-47052R3

Dear Dr. Pickering,

We’re pleased to inform you that your manuscript has been judged scientifically suitable for publication and will be formally accepted for publication once it meets all outstanding technical requirements.

An invoice will be generated when your article is formally accepted. Please note, if your institution has a publishing partnership with PLOS and your article meets the relevant criteria, all or part of your publication costs will be covered. Please make sure your user information is up-to-date by logging into Editorial Manager at Editorial Manager® and clicking the ‘Update My Information’ link at the top of the page. For questions related to billing, please contact billing support.

Kind regards,

Shabnam Iezadi, Ph.D.

Academic Editor

PLOS One

Additional Editor Comments (optional):

Reviewers’ comments:

---

## [Editor Report · Acceptance letter]

PONE-D-25-47052R3

PLOS One

Dear Dr. Pickering,

I’m pleased to inform you that your manuscript has been deemed suitable for publication in PLOS One. Congratulations! Your manuscript is now being handed over to our production team.

Lastly, if your institution or institutions have a press office, please let them know about your upcoming paper now to help maximize its impact. If they’ll be preparing press materials, please inform our press team within the next 48 hours. Your manuscript will remain under strict press embargo until 2 pm Eastern Time on the date of publication. For more information, please contact onepress@plos.org.

Kind regards,

on behalf of

Dr. Shabnam Iezadi

Academic Editor

PLOS One